# Diffusion in dense supercritical methane from quasi-elastic neutron scattering measurements

Umbertoluca Ranieri[1,2✉], Stefan Klotz [3], Richard Gaal[4], Michael Marek Koza[2] & Livia E. Bove [3,5✉]

Methane, the principal component of natural gas, is an important energy source and raw material for chemical reactions. It also plays a significant role in planetary physics, being one of the major constituents of giant planets. Here, we report measurements of the molecular self-diffusion coefficient of dense supercritical $CH_4$ reaching the freezing pressure. We find that the high-pressure behaviour of the self-diffusion coefficient measured by quasi-elastic neutron scattering at 300 K departs from that expected for a dense fluid of hard spheres and suggests a density-dependent molecular diameter. Breakdown of the Stokes–Einstein–Sutherland relation is observed and the experimental results suggest the existence of another scaling between self-diffusion coefficient $D$ and shear viscosity $\eta$, in such a way that $D\eta/\rho$=constant at constant temperature, with $\rho$ the density. These findings underpin the lack of a simple model for dense fluids including the pressure dependence of their transport properties.

[1] Center for High Pressure Science & Technology Advanced Research (HPSTAR), Shanghai, China. [2] Institut Laue-Langevin, Grenoble Cedex 9, France. [3] Sorbonne Université, UMR CNRS 7590, Institut de Minéralogie, de Physique des Matériaux et de Cosmochimie (IMPMC), Paris, France. [4] Laboratory of Quantum Magnetism (LQM), École Polytechnique Fédérale de Lausanne, Lausanne, Switzerland. [5] Dipartimento di Fisica, Universitá di Roma La Sapienza, Roma, Italy. ✉email: umbertoluca.ranieri@hpstar.ac.cn; liviaeleonora.bove@uniroma1.it

M ethane is one of the simplest molecular fluids and the most abundant organic molecule in the Universe. On account of this, it represents a model system for more complex organic molecules, an important feedstock for energy applications, and a major constituent of planetary interiors[1,2]. For example, methane is abundant on Titan where it plays the role of water on Earth, with liquid surface reservoirs, yellow clouds, and rain[3]. Since its discovery, the presence of methane on Mars has evoked visions of life on that planet[4]. The properties of methane at high pressure $P$ and high temperature $T$ are crucial for understanding the stability and formation of reduced hydrocarbons in the Earth's mantle[5], and for describing the geodynamics of giant planets such as Uranus and Neptune[6,7].

Methane is a non-polar, highly symmetric tetrahedral molecule with low polarizability. Therefore, the effective pair potential between molecules has negligible three-body terms and negligible dependence on the thermodynamic state[8]. Methane is isoelectronic with water, but it displays profoundly different physical properties due to its quasi-sphericity and to the lack of directional hydrogen bonds. For this reason, experimental findings for methane can be easily compared to theoretical models and computational results. At room temperature, methane crystallizes upon compression at ~1.4 GPa into the so-called phase I, where C atoms occupy fcc lattice sites and H atoms are free to rotate around them[9]. The application of higher pressures has many remarkable effects, such as transition to other crystalline structures[9–12], distortion of the tetrahedrality with consequent rise of a non-zero dipole moment[13], and reaction with other simple molecules to form $CH_4$–$H_2$ and $CH_4$–$H_2O$ inclusion compounds[13,14], for example. Solid methane has been studied under high pressures of hundreds of GPa and high temperatures of thousands of Kelvin[9–12,15]. However, few studies exist on room-temperature fluid methane at pressures from a fraction of GPa to 1.4 GPa. The single-particle rotational and translational diffusion coefficients, which are among the main transport properties of a fluid, are still unknown under those conditions. This is because the quasi-elastic neutron scattering (QENS) and nuclear magnetic resonance (NMR) techniques were limited to pressures of the order of a fraction of GPa in the past.

Previous QENS measurements reported values of the rotational and translational diffusion coefficients up to a maximum $P$ of 0.4 GPa[16]. The translational diffusion coefficient (or self-diffusion coefficient) has also been measured by NMR at room temperature up to 0.2 GPa[17–19], and along several low-$T$[17,18,20] and high-$T$[17,19] isotherms up to comparable pressures. At the lowest investigated temperatures (110 and 140 K), when the system is a liquid, the entire isotherms up to the freezing pressures could be covered[20]; however, at higher temperatures, higher pressures are required to reach the freezing point. Melting[21] and viscosity data[22] indirectly suggested that supercritical methane at room temperature could exhibit a "locking" of the rotational motion at pressures above 0.8 GPa, but this hypothesis could not be verified.

In this work, we report QENS measurements of the self-diffusion coefficient of dense supercritical methane at the constant temperature of 300 K, and pressures between 0.12 and 1.44 GPa. The measurements have been carried out at the IN6-SHARP spectrometer installed at the Institut Laue-Langevin (ILL) in Grenoble, France. QENS is a well-suited technique to probe single-particle dynamics of atoms and molecules in fields from materials science to biology, and in particular for hydrogenated samples. Our experimental results are compared with published computational values for the dense hard-sphere fluid, and with the prediction of the well-known phenomenological relation linking the self-diffusion coefficient to the shear viscosity, namely the Stokes–Einstein–Sutherland equation.

## Results

**Investigated pressure range.** Two high-pressure setups were employed to cover a wide $P$ range between 0.12 GPa and the freezing pressure. High-quality data over the $P$ range from 0.12 to 0.50 GPa were collected with a continuously loaded pressure cell, using methane both as the sample and the pressure transmitting medium. A Paris–Edinburgh press was employed to collect data up to 1.44 GPa, at which pressure freezing was not observed. Details about the high-pressure setups are given in the "Methods" section. The critical point of methane is at 191 K and 4.6 MPa, so the $T$-$P$ range investigated by the present experiment is entirely in the supercritical region, far from the critical point and also far from the maxima/minima in certain thermophysical properties associated with the so-called Widom line[23,24]. The melting curve given in ref. [25] passes through 300 K at 1.38 ± 0.02 GPa and our highest investigated pressure was 1.44 ± 0.05 GPa, which is slightly above the freezing pressure, meaning that the sample could have possibly been in the metastable fluid region.

At the lowest investigated pressure (0.12 GPa) and 300 K, the density is 0.3582 g cm$^{-3}$, which is more than twice the density at the critical point (0.1627 g cm$^{-3}$). The covered $P$ range corresponds to a range of density variation of 70% of the initial value, from 0.3582 g cm$^{-3}$ at 0.12 GPa to 0.6161 g cm$^{-3}$ at 1.44 GPa. The second value is considerably greater than the density of liquid methane at ambient pressure (for example, 0.4389 g cm$^{-3}$ at 100 K) and is also greater than the density of liquid methane at any pressure. Density values given throughout this paper are taken from the equation of state of ref. [26] up to 1 GPa and from its extrapolation above. Above ~0.7 GPa, methane at room temperature has a greater shear viscosity than the higher shear viscosity that can be found in liquid methane (~250 × 10$^{-6}$ Pa s, along the low-$T$ melting line[27]).

**Fitting model, obtained diffusion coefficient, and comparison with the literature data.** Figure 1 depicts typical empty cell-subtracted spectra of methane at selected $P$ and $Q$ values, $Q$ being the modulus of the wavevector transfer. The (quasi-elastic) signal gets broader with decreasing $P$ or with increasing $Q$, as expected in case of translational motion, and is correctly reproduced by a Lorentzian function. The rotational motion was ignored in the fitting model, similarly to our previous QENS study of methane diffusion at the interface of ice clathrate structures at 0.8 GPa[28]. This choice is discussed in Supplementary Note 1. No systematic deviations were observed during the fitting of the present data that might indicate the presence of a significant contribution from the rotational motion. Best Lorentzian fits to the spectra are reported in Fig. 1 and can be seen to accurately describe the measured signal with no systematic residuals.

Figure 2 depicts the half width at half maximum $\Gamma(Q)$ of the Lorentzians as a function of $Q^2$, at each investigated pressure. $\Gamma(Q)$ is observed to increase linearly at small $Q$ and to partially saturate at high $Q$. At small momentum transfers, i.e., when one is looking at the long distances, the translational motion is well approximated by Fickian diffusion, for which $\Gamma(Q) = \hbar D Q^2$, with $D$ the self-diffusion (or translational diffusion) coefficient. At high momentum transfers, i.e., when one is looking at shorter distances, the microscopic details of the translational motion become relevant and the $Q^2$ dependence of $\Gamma(Q)$ can show considerable deviations from linearity[29]. The model that was found to correctly describe the $Q$ dependence of $\Gamma(Q)$ in the present data is the translational random-jump diffusion model originally proposed by Singwi and Sjölander for liquid water[30], for which:

$$\Gamma(Q) = \frac{\hbar D Q^2}{1 + \tau D Q^2}, \quad (1)$$

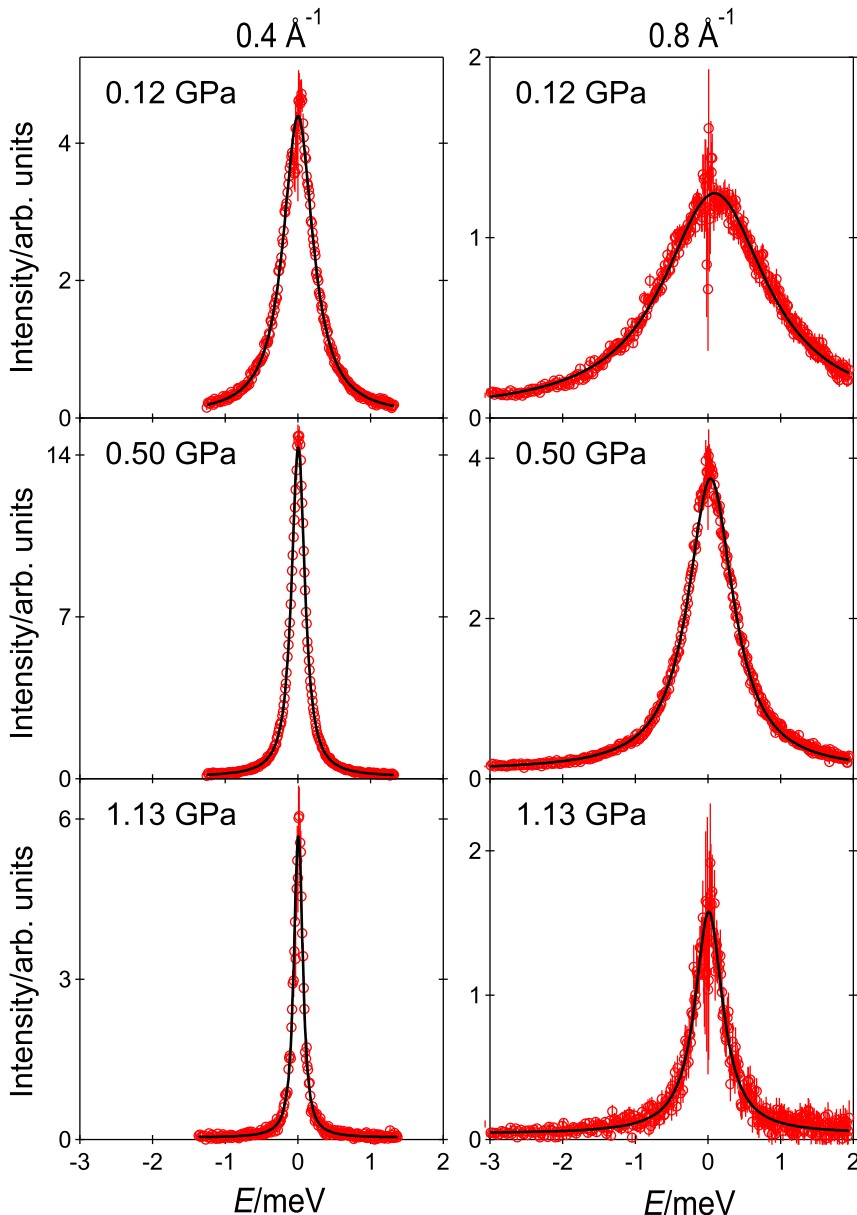

**Fig. 1 Examples of measured spectra.** QENS spectra of methane at 300 K and the indicated pressure and wavevector transfer values. Experimental data (empty circles) are compared to the resolution-corrected best Lorentzian fits (solid lines). Error bars were calculated by the square root of absolute neutron count combined with the law of propagation of errors.

where $\tau$ represents the time spent by the diffusing particle at quasi-equilibrium sites between rapid jumps. The same model was employed in our QENS work on interfacial methane[28] and previously in a QENS study of the diffusion of methane in microporous silica[31]. QENS data of liquid water over wide ranges of temperature and pressure have also been analysed using the Singwi–Sjölander formula[29,32–35]. Best fits of $\Gamma(Q)$ using Eq. (1) are reported in Fig. 2. More details about the data analysis are given in the "Methods" section.

The obtained self-diffusion coefficient $D$ is plotted as a function of pressure in Fig. 3. Values range from $22.59 \times 10^{-9}\,\mathrm{m^2\,s^{-1}}$ at 0.12 GPa to $3.69 \times 10^{-9}\,\mathrm{m^2\,s^{-1}}$ at 1.44 GPa. The values obtained for the parameter $\tau$ are tabulated in Supplementary Table 1 and commented in Supplementary Note 2. For comparison, $D = 23200 \times 10^{-9}\,\mathrm{m^2\,s^{-1}}$ in ambient-pressure, room-temperature gaseous methane[36], when its density is only $0.00066\,\mathrm{g\,cm^{-3}}$ [26]. Values of 3.61 and $3.6 \times 10^{-9}\,\mathrm{m^2\,s^{-1}}$ were reported for liquid methane at 0.0344 MPa

and 100.0 K (along the liquid-vapor coexistence line)[18] and at 0.15 MPa and 102 K[37], respectively. The self-diffusion coefficient was found to be roughly constant in liquid methane along the freezing (melting) line between 91 and 140 K[18,20], but it has not been possible to check if it is comparable at higher temperatures along the melting. The only measurements that are available along the melting line in the literature are: $2.52 \times 10^{-9}\,\mathrm{m^2\,s^{-1}}$ at 0.0117 MPa and 91 K (the triple point)[18], $2.62 \times 10^{-9}\,\mathrm{m^2\,s^{-1}}$ at 0.08 GPa and 110 K[20], and $2.85 \times 10^{-9}\,\mathrm{m^2\,s^{-1}}$ at 0.22 GPa and 140 K[20]. Our results show that in supercritical methane at 300 K and the freezing pressure, the self-diffusion coefficient is significantly larger than in liquid methane along the low-$T$ melting line.

A good fit to our values of $D$ over the full investigated $P$ range is given by the following empirical relation:

$$D = \exp\left(3.5495 - 4.3717P + 4.1527P^2 - 1.5306P^3\right), \quad (2)$$

where $D$ is in units of $10^{-9}\,\mathrm{m^2\,s^{-1}}$ and $P$ in GPa. Equation (2)

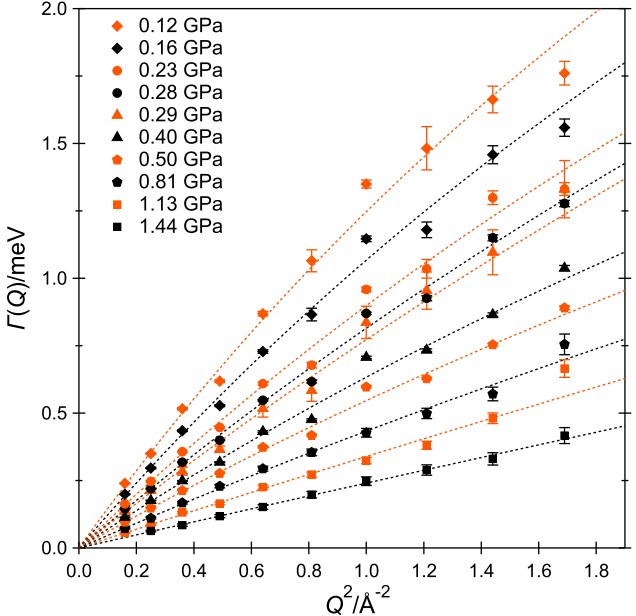

**Fig. 2 Wavevector transfer Q dependence of the QENS signal.** Half-width at half maximum $\Gamma(Q)$ of the Lorentzian fits of the quasi-elastic signal as a function of $Q^2$. Error bars correspond to one standard deviation, as obtained from fits such as those of Fig. 1. The best fits to the data using Eq. (1) are shown as dashed lines.

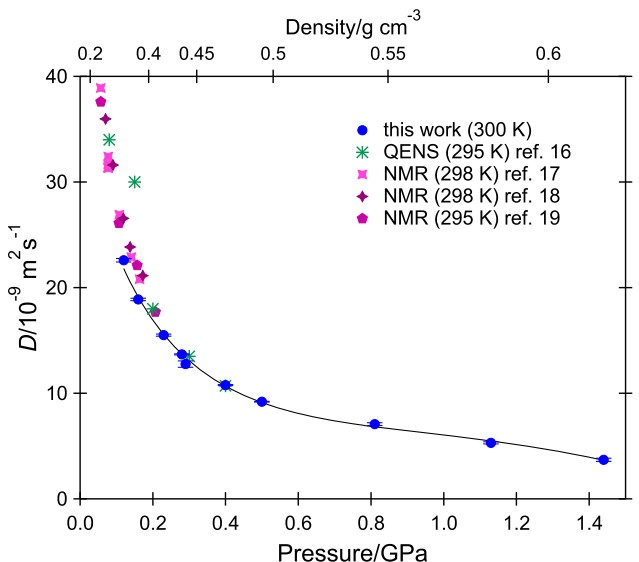

**Fig. 3 Pressure dependence of the diffusion coefficient.** Self-diffusion coefficient $D$ of methane at 300 K as a function of pressure and its best fit (solid line), reported in Eq. (2). Error bars correspond to one standard deviation, as obtained from the fits of ten data points shown in Fig. 2. Room-temperature literature values[16–19] above 0.05 GPa are also shown. The upper horizontal axis reports the density, as obtained from the equation of state of methane at 300 K.

accurately reproduces the measured $P$ dependence of $D$ (the standard deviation is 3.3%) and can be used for interpolations. In Fig. 3, we also compare our results with those available in refs. [16–19] for room-temperature methane. The previous QENS study[16] covered a pressure range up to 0.4 GPa at 295 K and reported greater $D$ values than the present study, except at 0.4 GPa, for which an almost identical value was given. There are three sets of NMR data along the room-temperature isotherm:

up to 0.16 GPa at 298 K by Harris[17], up to 0.17 GPa at 298 K by Oosting and Trappeniers[18], and up to 0.207 GPa at 295 K by Greiner–Schmid and co-workers[19]. Their respective precisions in $D$ were estimated to be ±2, ±2, and ±5%. Over the narrow common pressure range, our results follow the same pressure trend as the NMR values but are 10–15% smaller (see Fig. 3), which can be considered to be satisfactory given the different techniques. A larger (20–25%) discrepancy was remarked at moderate pressures (up to 0.025 GPa) when comparing results from the NMR[17] and the tracer[38] techniques. High-pressure literature measurements of $D$ exist at temperatures from 91 to 454 K and pressures up to a maximum of 0.207 GPa by NMR[17–20] and QENS[16,39]. Supplementary Fig. 1 locates them in the $T$-$P$ phase diagram of methane. A molecular dynamics simulation study[40] of methane reported values for $D$ of 19.6 and $23.4 \times 10^{-9}$ $m^2 s^{-1}$ at 0.107 GPa and 295 K (not shown in Fig. 3), depending on the choice of the potential model. The second value, which was obtained using a Lennard–Jones potential with electrostatic interactions, compares very well with our experimental value of $22.59 \times 10^{-9}$ $m^2 s^{-1}$ at 0.12 GPa. Finally, in Supplementary Fig. 2, we compare the pressure dependence of our data with results of a molecular dynamics simulation study of the Lennard–Jones fluid[41]. Supplementary Fig. 2 reports the logarithm of $D$ as a function of $P$ in terms of reduced quantities as defined in the ref. [41]. The model is not able to catch the trend of the experimental data.

**Hard-sphere model.** The hard-sphere model is a useful model for calculating the transport properties of dense fluids, because these properties are largely determined by the repulsive part of the intermolecular potential for which the hard-sphere interaction is the simplest representation. The self-diffusion coefficient of liquid and supercritical methane up to ~0.2 GPa[17,20] was initially shown to be globally in good agreement with computational results from ref. [42] for a fluid of hard spheres that are perfectly smooth, meaning that the translation-rotation coupling is negligible. Other computational results also gave a good fit to the experimental self-diffusion data for methane at low and room temperature leading to the conclusion that methane could be adopted as a model smooth hard-sphere fluid with no translation-rotation coupling as far as transport properties were concerned[43]. However, later, new data and simulations indicated that methane shows some translation-rotation coupling at low and high temperatures[19,44]. The effect of the translation-rotation coupling can be taken into account by analogy with a rough hard-sphere fluid[45]. This phenomenological model, which is valid between twice the critical density and the freezing density of the hard-sphere fluid, describes the lowering of the self-diffusion coefficient due to the translational-rotational coupling in an averaged way, and assumes that the experimental self-diffusion coefficient $D_{exp}$ is:

$$D_{exp} = D_{rhs} = A D_{shs}(\sigma), \qquad (3)$$

where $D_{rhs}$ and $D_{shs}(\sigma)$ are the self-diffusion coefficients of the rough and smooth hard-sphere fluids, respectively, $A$ is a coupling parameter smaller than unity and $\sigma$ is the sphere diameter[45]. $D_{shs}(\sigma)$ is meant to be taken from molecular dynamics simulations while $A$ and $\sigma$ are free parameters, which can be adjusted to match the density dependence of $D_{rhs}$ with that of $D_{exp}$. Both parameters should be independent of temperature and density by definition, but in practice $\sigma$ is often permitted to be temperature (though not density) dependent. Greiner-Schmid and co-workers[19] compared their experimental data to the computational results of ref. [46] with a $T$-dependent coupling parameter $A$, and found values ranging between 0.95 and 1.07 with no clear

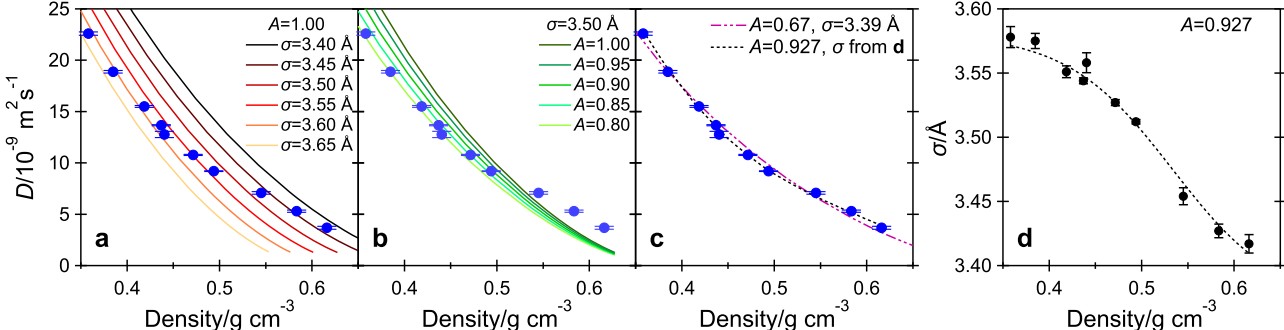

**Fig. 4 Comparison with the molecular dynamics simulations of the hard-sphere fluid. a–c** Self-diffusion coefficient $D$ from our measurements of methane (full circles) and from the simulation results for the hard-sphere fluid reported in ref. [48] (lines), as a function of density. Simulated values are reported for: **a** $A = 1.00$ and six different values of the hard-sphere diameter $\sigma$, **b** $\sigma = 3.5$ Å and five different values of the translation-rotation coupling parameter $A$, and **c** $A = 0.67$ and $\sigma = 3.39$ Å, $A = 0.927$ and $\sigma$ from the fit of **d**. **d** $\sigma$ values for which Eq. (3) is satisfied with $A = 0.927$, as a function of density, and the best sigmoidal fit (dashed line). Error bars correspond to one standard deviation.

trend with $T$. Harris[44] re-analysed the data of ref. [19] using more recent computational results[47] and a $T$-independent coupling parameter, and found $A = 0.927$ over the range 295–427 K. He also re-analysed the data of refs. [17,20]; he found $A = 0.90$ for the range 110–323 K[44].

In Fig. 4a, we compare our experimental values of $D$ with recent computational results[48] for the dense fluid of smooth hard spheres from molecular dynamics calculations carried out with large system sizes. Ref. [48] reports the self-diffusion coefficient in units of $\sigma(k_B T/m)^{1/2}$ as a function of dimensionless reduced number density $n\sigma^3$, where $k_B$ is the Boltzmann constant, $m$ is the molecular mass, and $n = \rho N_A / M$ is the number density, with $\rho$ the mass density, $N_A$ the Avogadro number, and $M$ the molar mass. In Fig. 4a, the simulated self-diffusion coefficient was converted into the unit of the present work for six selected values of $\sigma$ and $T = 300$ K. Reduced number density was converted into mass density. In Fig. 4a, we directly report results of ref. [48] for smooth hard spheres so this approach corresponds to assuming $A = 1$. One can see from the figure that the experimental $D$ coincides with the computational one for $\sigma \sim 3.62$ Å at our two lowest investigated densities. As $\rho$ increases, the experimental $D$ corresponds to the diffusion coefficient obtained for a hard-sphere fluid with decreasing values of $\sigma$ and eventually for $\sim 3.43$ Å at our two highest densities before crystallization. In other words, the experimental results do not follow the results simulated for a smooth hard-sphere fluid with a constant ($\rho$-independent) sphere diameter $\sigma$. A small deviation from the smooth hard-sphere prediction has been previously observed in compressed liquid methane at low $T$ for the few highest pressure points available in ref. [20]. However, the present study is the first clear observation of such behavior for methane, to the best of our knowledge. In ref. [20] data from six isotherms from 110 to 323 K could be superposed when the reduced self-diffusion coefficient was plotted against reduced number density through the choice of a $T$-dependent, $\rho$-independent diameter $\sigma$. The present data cannot be superposed to that common curve for any density-independent $\sigma$. This approach is discussed further in Supplementary Note 3 and the reduced diffusion coefficient is plotted in Supplementary Fig. 3.

In Fig. 4b, our values of $D$ are compared with those simulated in ref. [48] for the hard-sphere fluid for $\sigma = 3.5$ Å (the value found in ref. [44] for the room-temperature data of ref. [19]) and $T = 300$ K after multiplication by the coupling parameter $A$, for five different values of $A$ between 0.8 and 1.0. As can be seen in the figure, the computational results for $A = 0.85$ reproduce the experimental $D$ within 7% up to $\sim 0.5$ g cm$^{-3}$, but the model predicts considerably smaller self-diffusion coefficients for any value of $A$ at higher

densities. Our data can be reasonably well fitted with $\sigma = 3.39$ Å and $A = 0.67$ over their entire range of pressures (see Fig. 4c), but these values are by far too small compared to the literature low-$P$ results ($\sigma = 3.47$–$3.50$ Å and $A = 0.90$–$0.93$[44]). One must conclude that, despite the correction accounting for the translation-rotation coupling, the hard-sphere model does not correctly describe the experimental results for any pair of reasonable constant values of $\sigma$ and $A$. In order to match the computational results with our experimental results, one has to relax at least one of the two assumptions that $\sigma$ and $A$ are density independent. Since there is reason to believe that $A$ does not increase with increasing $P$[21,22], we suggest that $A$ should be kept constant and equal to the low-$P$ value while the diameter should be allowed to change (decrease) with density. We compared our values of $D$ with the computational results[48] after multiplication by $A = 0.927$ (the value found in ref. [44] for the data of ref. [19]) and, at each investigated density point, we matched $0.927 D_{shs}(\sigma)$ with $D_{exp}$ by adjusting the hard-sphere diameter $\sigma$. The values of the density-dependent equivalent diameter $\sigma$ so obtained are reported in Fig. 4d with their best fit. They vary from 3.58 Å at our lowest densities to 3.42 Å at our highest density. Finally, for $A = 0.927$ and the $\rho$-dependent $\sigma$ of Fig. 4d, the computational values of $D$ are those plotted in Fig. 4c, where they can be seen to agree well with the experimental ones. If $A$ decreases with pressure, the actual variation of $\sigma$ should be even larger than that reported in Fig. 4d.

**Stokes–Einstein–Sutherland and fractional Stokes–Einstein–Sutherland equations.** It is common to compare the pressure dependence of the microscopic self-diffusion coefficient with that of the macroscopic shear viscosity $\eta$ using the Stokes–Einstein–Sutherland (SES) equation, which predicts:

$$\frac{D}{T} = \frac{k_B}{\pi \sigma C} \frac{1}{\eta}, \tag{4}$$

with $\sigma$ an effective (hydrodynamic) diameter and $C$ a constant depending on the geometry of the motion and comprised between 2 (slip limit) and 3 (stick limit). Although the SES relation strictly only applies to macroscopic spherical particles diffusing in a solvent, it has been found to work remarkably well on the molecular scale for the self diffusion of atomic and molecular liquids, in which case the solute particle is the same as the solvent particles, with some flexibility in the assignment of $\sigma$ and $C$. When the SES relation fails, the deviation can be generally interpreted as being due to the fact that $\sigma$ and/or $C$ depend on the thermodynamic conditions. Vice versa, if $\sigma$ is known to depend on the thermodynamic conditions (as we

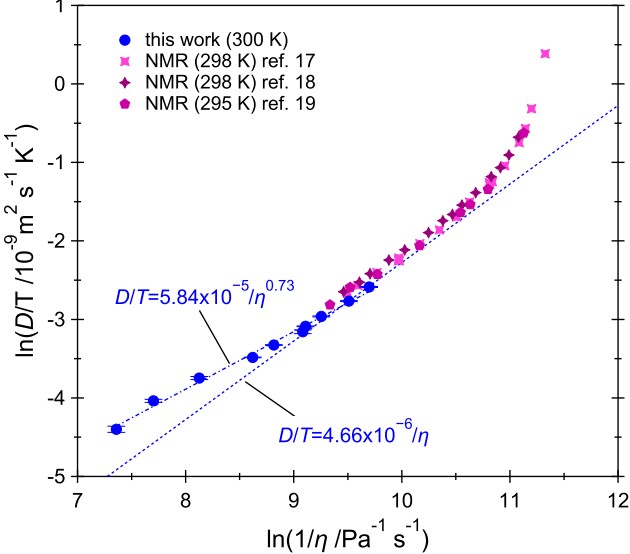

**Fig. 5 Test of the Stokes-Einstein-Sutherland and fractional Stokes-Einstein-Sutherland relations.** $\ln(D/T)$ as a function of $\ln(1/\eta)$, with self-diffusion coefficient values from this work and from refs. [17-19] and viscosity values from refs. [22,50]. Error bars correspond to one standard deviation. The best fit to our data using a fractional Stokes–Einstein–Sutherland relation ($D/T = K/\eta^\xi$) is shown as dash-dotted line and the values of $K$ and $\xi$ are indicated. The Stokes–Einstein–Sutherland relation (dashed line) predicts $D/T = K/\eta$.

have shown in the previous subsection), then the SES relation is expected to fail. Very often, the SES relation can be replaced by a "fractional" Stokes–Einstein–Sutherland (FSES) equation, which predicts:

$$\frac{D}{T} \propto \frac{1}{\eta^\xi}, \qquad (5)$$

where $0 \leq \xi \leq 1$ is usually independent of both $P$ and $T$[49]. Harris[49] examined the literature database available for the self-diffusion coefficients and viscosities of molecular and ionic liquids at modest pressures, and concluded that the FSES equation holds over wide $T$ ranges with fitted values of $\xi$ between 0.79 and 1.00.

Figure 5 depicts $\ln(D/T)$ as a function of $\ln(1/\eta)$ using our values of $D$ as well as the room-temperature values of refs. [17-19] and $\eta$ values interpolated from refs. [22,50]. The viscosity of supercritical methane was measured up to 1 GPa at 298 K in ref. [22] and up to 1.32 GPa at 293–295 K in ref. [50]; therefore, a small extrapolation to 1.44 GPa was needed. In Fig. 5, the SES relation ($D/T = K/\eta$, where $K$ is a constant) simply appears as a straight line with slope 1 and the FSES relation ($D/T = K/\eta^\xi$) appears as a straight line with slope $\xi$. As can be seen in the figure, a FSES relation with an exponent $\xi = 0.73 \pm 0.02$ successfully describes our data over their entire range of pressures (0.12–1.44 GPa). The maximum deviation of our data points from the best fit is 7% and the standard deviation is 5.1%. Globally, the fit is believed to be satisfactory given the extended range of pressure variation and the error bars on $D$ and $P$. On the other hand, the slope of $\ln(D/T)$ versus $\ln(1/\eta)$ is strongly pressure-dependent in the literature self-diffusion data[17-19], which cover the pressure range from 4.6 MPa (the critical pressure) to 0.2 GPa. It is larger than 1 below 0.04 GPa and smaller than 1 above 0.04 GPa (see Fig. 5), meaning that the SES relation is violated in both regimes but it holds at ~0.04 GPa. This can be also appreciated by looking at the pressure dependence of $D\eta$, which is reported in Supplementary Fig. 4. At constant $T$, the SES relation predicts a constant product

between $D$ and $\eta$. Combination of our results with the literature data[17-19] unambiguously shows that the calculated product $D\eta$ for methane along the room-temperature isotherm decreases sharply with $P$ below 0.04 GPa, has a minimum at ~0.04 GPa, and increases slowly with $P$ above 0.04 GPa (see Supplementary Fig. 4). Similarly, in the Lennard–Jones fluid, $D\eta$ has a minimum as a function of the temperature along isochores[51]. It should be mentioned that the SES relation is not expected to hold below twice the critical density, i.e., below ~0.08 GPa at room $T$. Some considerations about the hydrodynamic diameters and parameters $C$ satisfying Eq. (4) along our isotherm are reported in Supplementary Note 4. Supplementary Fig. 5 shows that the SES relation holds in compressed liquid methane along the 110, 140, and 160 K isotherms.

Finally, another simple scaling happens to describe our experimental results: the factor $D\eta/\rho$ is constant with maximum deviations of ±10% over the $P$ range investigated here (both $D\eta$ and $\rho$ increase by ~70%). Such a scaling has been first suggested by Dullien[52] to work for a set of liquids, including methane at very modest pressures. The pressure dependence of $D\eta/\rho$ is reported in Supplementary Fig. 6 for our values of $D$ as well as the room-temperature values of refs. [17-19]. The factor $D\eta/\rho$ decreases fast with pressure for $P$ up to ~0.2 GPa at room temperature (see Supplementary Fig. 6) and decreases slowly with pressure along the 110, 140, and 160 K isotherms (see Supplementary Fig. 7). Should it be confirmed by future studies that the factor $D\eta/\rho$ is constant for dense supercritical methane, it will be possible to use it for extrapolations along the isotherms for which $\eta$ is known, and $D$ is only available at low pressure in the literature (this is presently the case between 327 and 454 K[19,50]). In a similar way, the product $D\rho$ is found[53] to be constant from ambient to the critical $P$ (consistently with the zero-order approximation of the binary-collision expansion of $D$ in terms of the density[18]), which is useful for extrapolations at those moderate pressures. $D\rho$ drops upon isothermal compression above the critical $P$[54]. More sophisticated models for predicting the self-diffusion coefficient (and viscosity) of fluids have been developed over the last four decades[55–57]; reviewing them falls beyond the scope of the present work. The popular simple relation $D\eta/\rho^{1/3}$ = constant, which is a modification of the Stokes–Einstein–Sutherland relation without a hydrodynamic diameter[51], does not describe well our experimental results along the 300 K isotherm.

## Discussion
In the present study, we compared the experimental self-diffusion coefficient of methane along the 300 K isotherm with published computational values for a dense smooth hard-sphere fluid[48], and found poor agreement for any density-independent sphere diameter. Similar deviations, with the experimental self-diffusion coefficient being higher than that predicted based on the smooth hard-sphere model, were also reported for more complex molecular liquids approaching their freezing pressures, for example $CCl_4$[45], $C_6D_6$, and $Si(CH_3)_4$[58]. These observations are examples of limitations of the model when tested over wide ranges of pressure variation. The hard-sphere model is often tested in a region of the phase diagram of fluid systems, where the effects of the repulsive interaction among molecules are averaged by the distances. When the molecules are forced close together and the repulsive part of the potential is probed in higher details, more accurate models are needed to describe the dense fluid. A better agreement with our experimental results is found when the coupling of the translational and rotational motions is taken into account through the correction suggested in ref. [45] but still, the experimental values lie well above the computational results at pressures above ~0.5 GPa.

We concluded that the equivalent hard-sphere diameter of methane must be reduced upon compression along the 300 K isotherm. Possible reasons for this might be the softness of the intermolecular potential and/or orientational ordering arising at high density, both neglected in the rough hard-sphere model. It was similarly suggested that the effective hard-core diameters of $C_6D_6$ and $Si(CH_3)_4$ become a function of density at high packing fractions[58]. The good quality of our data allows us to quantify this effect. Values of the diameter were estimated assuming a constant translation-rotation coupling, and a significant decrease from 3.58 to 3.42 Å was found with increasing density. It is interesting to note that the equivalent hard-sphere diameter is not found to decrease linearly with density, but decreases faster in the high-density regime, where rotational locking was inferred from previous melting and viscosity data[21,22], as mentioned in the introduction. The more recent viscosity measurements[50] confirmed that the viscosity increases with $P$ more rapidly than expected above 0.8 GPa at room temperature, but did not show a radical divergence in viscosity along the investigated high-$T$ isotherms (up to 6 GPa and 673 K). Thus, the scenario of a locking of the rotational motion remains to be corroborated. Orientational ordering would enable the system to get more tightly packed, thus reducing the equivalent single-molecule size as observed in the present study. Furthermore, at the highest pressures, enhanced orientational correlations would make the motion of adjacent molecules occurring without significant overlap between core repulsion regions, which would also imply larger values of the diffusion coefficient than those predicted by the rough hard-sphere model. Further investigations, including computational studies, are needed to confirm these hypotheses. Interestingly, if orientational ordering were confirmed, the rotation of the molecules would be more hindered in the fluid phase close to the freezing pressure than in the crystal.

We also tested the validity of the Stokes–Einstein–Sutherland equation against our findings for dense supercritical methane. To our knowledge, the Stokes–Einstein–Sutherland equation had never been tested experimentally on a supercritical fluid up to the freezing pressure. We found that the SES equation is violated in our data, as the product $D\eta$ increases by 70% from 0.12 to 1.44 GPa at the constant temperature of 300 K. A fractional Stokes–Einstein–Sutherland equation with an exponent $\xi = 0.73 \pm 0.02$ correctly represents our data over their entire range of pressures (0.12–1.44 GPa). This value of $\xi$ is close to the value (0.76) obtained for hot dense liquid water along the 400 K isotherm[32], and is slightly smaller than the bottom limit (0.79) of the range of values reported for molecular and ionic liquids at modest pressures and different temperatures[49]. Both the smooth hard-sphere fluid and the model Lennard–Jones fluid were predicted to follow FSES relations with considerably larger values of $\xi$ of 0.96[48] and 0.92[49], respectively. It is worth emphasizing that $D\eta$ has a shallow minimum as a function of pressure at ~0.04 GPa. The minimum might also exists in other systems and might explain why the SES relation appears to hold in part of the thermodynamic phase diagrams.

Breakdown of the SES equation is common in the literature of supercooled and glass-forming liquids, including supercooled water[59]. The failure of the Stokes–Einstein–Sutherland relation in the supercooled regime has been addressed by various theoretical perspectives[60,61]; it is often attributed to the presence of dynamical heterogeneity[62,63] and its quasi-universality has been recently explained by isomorph theory[64]. However, the case of a dense fluid subjected to a strong density variation upon isothermal compression is very different and to our knowledge has so far not been considered theoretically. We hope that our study will motivate the development of theories for explaining the breakdown of the SES relation observed here in dense supercritical methane.

To conclude, we found that all of the popular models tested here ($D$ = constant along the melting line, the Lennard–Jones model, the smooth and rough hard-sphere models, the Stokes–Einstein–Sutherland equation, and the fractional Stokes–Einstein–Sutherland equation with the power–law exponent of the Lennard–Jones fluid) are inadequate to reproduce the measured high-pressure behavior of the self-diffusion coefficient of methane along the room-temperature isotherm. This highlights the lack of a simple model for predicting the single-particle dynamics in dense fluid methane, and in molecular fluids at high densities in general. A fractional Stokes–Einstein–Sutherland equation describes well our data for an exponent (0.73) smaller than the typical values reported[49] for simple fluids. The factor $D\eta/\rho$ remains fairly constant at constant temperature over the pressure range investigated here, and it would be very interesting to check if this empirical scaling holds along other isotherms.

The present study might be extended to fluid methane along lower-$T$ isotherms in the vicinity of the critical point as well as along higher-$T$ isotherms, where higher pressures and densities can be achieved. There is also a need for more high-pressure viscosity data of methane at temperatures below 273 K, the current limit being 0.031 GPa in the experimental literature dataset[65] and 0.2 GPa from the equation of state-like viscosity model of ref. [27]. In general, data on the properties of compressed supercritical fluids are surprisingly scarce, despite the increasing number of applications of supercritical fluids in extraction and purification technologies, for example in food, petrochemical, nuclear waste, and pharmaceutical industries. The solubility in supercritical fluids depends on density and diffusivity. A better knowledge of the diffusivity of supercritical fluids would allow for an optimization of their dissolving ability by tuning pressure and temperature. We believe that technology would benefit from such a guidance. We must acknowledge that significant progresses have been made in recent years concerning the fundamental problem of dividing the phase diagram into separate regions, where a supercritical fluid possesses liquid-like and gas-like properties[23,24]. There is still a lot to do on compressed fluid binary mixtures. Pressures of the order of a fraction of GPa are increasingly accessible by the industry and are also relevant for Earth and planetary science. As an example, methane-rich hydrous fluids are ubiquitous in the Earth, where they exist down to upper mantle conditions, and are believed to have a strong impact on elastic waves propagation at depth[66].

## Methods

**High-pressure setup**. For the measurements up to 0.5 GPa, we used a cylindrical pressure cell of aluminum alloy whose internal diameter was 6 mm. A cylindrical aluminum spacer of 5.5 mm in diameter was inserted to reduce the multiple scattering contributions and about 200 mm³ of methane was contained in the resulting hollow circular cylinder (thickness of 0.25 mm, height of about 40 mm). The $CH_4$ (purity > 99.95%) bottle was purchased from AirLiquide, Saint Priest, France. The sample pressure was changed using a methane-compatible gas compressor, and was directly measured by a manometer attached to the capillary connecting the compressor with the cell (no piston or separator was used; methane filled the sample chamber as well as the capillary). Fluctuations of this value around the desired pressure were always below 5 MPa. Spectra of methane at 300 ± 1 K were recorded at six pressures between 0.12 and 0.50 GPa: 0.12, 0.16, 0.23, 0.28, 0.40, and 0.50 GPa. The acquisition time was typically 3–4 h per $P$ point. After the measurements of methane, the sample was evacuated and the empty cell was measured at the same experimental conditions.

Additional data were collected using the VX5 Paris–Edinburgh press[67] in combination with a loading clamp[68]. In this setup, a quasi-spherical ~50 mm³ sample is encapsulated inside a metallic gasket and squeezed between two anvils. QENS measurements in the Paris–Edinburgh press have been previously performed by our group on liquid water[32–34] and solid methane clathrate hydrate[28]. For the present work, a setup similar to that of ref. [28] was employed: we used a copper-beryllium gasket and anvils made of zirconia-toughened alumina ceramics, described in the ref. [69]. The clamp module of the Paris–Edinburgh press

was inserted into a high-pressure vessel and methane was loaded at room temperature and 0.2 GPa. The employed gas compressor and high-pressure vessel, which are installed at the ILL, were described in ref. [68]. Approximately 30 mm$^3$ of $Na_2Ca_3Al_2F_{14}$ powder was loaded in the sample chamber to reduce multiple scattering and to serve as a pressure gauge.

Three sample loadings were performed in the Paris–Edinburgh press and for each of them, the gasket was sealed by applying a different load. Two samples were loaded by applying 60 and 115 kN and measured at IN6-SHARP at only one $P$ point each. In a second step, both samples were measured at the neutron diffractometer D20 at the ILL to determine their pressure from our equation of state of $Na_2Ca_3Al_2F_{14}$ (Birch–Murnaghan, $B_0$ = 62.8 GPa, $B_0'$ = 3.8, $V_0$ = 1080.7 Å$^3$). The sample at the lower pressure had a lattice parameter for $Na_2Ca_3Al_2F_{14}$ of 10.2463 ± 0.0010 Å, corresponding to $P$ = (0.29 ± 0.03) GPa. The sample at the higher pressure was found to be at 1.44 ± 0.05 GPa (lattice parameter of 10.187 ± 0.002 Å). The third sample was loaded by applying 85 kN and measured at IN6-SHARP, then compressed to 100 kN and measured again. These two values correspond to pressures of 0.81 ± 0.05 and 1.13 ± 0.05 GPa, by linearly interpolating the pressures obtained for the two previous samples. Upon further compression to 120 kN, the gasket failed and the sample was lost. The force was released completely and the empty cell was measured. All the measurements were performed at 300 ± 1 K. Acquisition times ranged between 6 and 12 h per pressure point.

**Data analysis**. IN6-SHARP is a time-of-flight spectrometer for cold neutrons with variable resolution. We set the wavelength of the incident neutrons to 5.12 Å, resulting in an instrumental energy resolution of about 0.08 meV (full width at half maximum) at zero energy transfer. Constant-$Q$ spectra from 0.4 to 1.7 Å$^{-1}$ with 0.1 Å$^{-1}$ steps were extracted for each $P$ point. Their intensity was normalized using the measurement of a vanadium standard and the empty-cell signal was subtracted using $P$-dependent and $Q$-dependent transmission values. Each spectrum (whose intensity is proportional to the self-dynamic structure factor $S_{inc}(Q, E)$) was fitted over a narrow quasi-elastic region using a Lorentzian function plus a flat background:

$$S_{inc}(Q, E) = \frac{I(Q)}{\pi} \frac{\Gamma(Q)}{(E - E_0)^2 + \Gamma^2(Q)} + B(Q). \quad (6)$$

Four free fitting parameters were used: area $I(Q)$ and half-width-half-maximum $\Gamma(Q)$ of the Lorentzian, zero shift $E_0$ of the energy-transfer axis, and flat background $B(Q)$. Fits were performed with weights inversely proportional to the error bars so that the scattered points close to $E = 0$ barely affect the fit results. Multiple scattering contributions to the spectra were neglected as the sample transmission was above 90% of the incident beam in both high-pressure setups. Stokes/anti-Stokes detailed balance was also ignored. Convolution with the instrumental resolution function was taken into account in the fits. The energy resolution function of the spectrometer was obtained by fitting the measurement of the vanadium standard with a pseudo-Voigt distribution; its width was found to vary from 0.076 meV at low $Q$ to 0.081 meV at high $Q$. We checked that the $Q$ dependence of $I(Q)$ accurately follows $\exp(-Q^2\langle u^2\rangle/3)$ at each pressure. The fitted values of $\langle u^2\rangle^{1/2}$ have no pressure dependence over the measured $P$ range within their error bars (typically 5%) and amount to 1.03 Å on average, which is within the range of values reported in the literature for liquid water at ambient conditions[70], for which many QENS studies exist. Finally, $\Gamma(Q)$ was fitted over the $Q$ range from 0.4 to 1.3 Å$^{-1}$ using the Singwi–Sjölander random-jump diffusion model reported in Eq. (1) to determine the two parameters $D$ and $\tau$ at each pressure. In the fits, the data points were weighted according to the estimated error in $\Gamma(Q)$. At the highest $P$ (1.44 GPa), the lowest $Q$ value (0.4 Å$^{-1}$) was not considered because $\Gamma(Q)$ was found to be smaller than the instrumental resolution, and thus is unreliable.

## Data availability
Data were generated at the ILL large-scale facility and are stored on the ILL data portal under https://doi.org/10.5291/ILL-DATA.CRG-2506.

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

## Acknowledgements

We acknowledge the "Fédération Française de Diffusion Neutronique" for provision of beam time on IN6-SHARP, and Claude Payre, James Maurice, Eddy Lelièvre-Berna, Jean Marc Zanotti, and Quentin Berrod for advice and technical assistance during the experiment. We also thank Thomas C. Hansen for help during the measurement on D20, and Jeppe C. Dyre, John Russo, and Francesco Sciortino for useful discussions.

## Author contributions

U.R., S.K., R.G. M.M.K., and L.E.B. performed the experiments. U.R. processed the experimental data, performed the analysis, drafted the manuscript and designed the figures. L.E.B. aided in interpreting the results and worked on the manuscript. All authors discussed the results and commented on the manuscript.

## Competing interests

The authors declare no competing interests.
