## [Peer Review File · Nature Communications]

REVIEWER COMMENTS

Reviewer #1 (Remarks to the Author):

Ranieri et al.

Diffusion in dense supercritical methane from quasi-elastic neutron scattering measurements

Nature Communications.

- What are the noteworthy results?

This work reports QENS measurements of the self-diffusion coefficient of methane in the supercritical region along a single isotherm, extending to the freezing line, that is to 1.44GPa. The work complements and extends the range of thermodynamic states of previous high pressure measurements, made mainly by spin echo NMR, for the liquid and dense gas states. The liquid state measurements also extend to the freezing line.

The data are analysed using several commonly used theoretical and empirical approaches: a) smooth and rough hard sphere theory, that is a comparison with literature molecular dynamics simulations for the hard-sphere fluid, whereby a temperature dependent diameter and an empirical “translational-rotational” coupling parameter can be assigned, b) the Stokes-Einstein-Sutherland equation relating the self-diffusion coefficient to the viscosity, which in this context is a comparison of the density or pressure dependence of the self-diffusion coefficient with that of the viscosity.

The authors conclude from the hard sphere comparison that:

a) Methane does not fit the smooth-hard sphere model, confirming what is known from earlier studies (ref. 41). The deviation from a linear dependence of D on the molar volume shown in Fig 3 is actually the norm for molecular fluids, being first reported by Parkhurst and Jonas in the 1970's (ref. 49). Though this is the first clear observation of such behaviour for methane, Harris and Trappeniers (ref 20) reported a similar trend for their highest pressure data (their Fig. 4). Van der Gulik has also reported similar behaviour for the viscosity of liquid CO₂ (maximum pressure 0.4 GPa) (P S van der Gulik, Physica A, 23, 81 (1997)).

b) Instead, the data can be fitted with the use of a translational-rotational coupling factor, but only with the use of a density dependent diameter. Again this is a result of the non-linear molar volume dependence, or, more precisely, the deviation from the smooth hard spheres volume dependence, which only approximates a linear molar volume dependence.

Such comparisons can only be made between twice the critical density, 0.32 g cm⁻³ for CH₄, (see ref. 39) and the freezing density of the hard sphere fluid, 0.94. The data presented do fit between these limits, but this point is omitted by the authors.

The Stokes-Einstein-Sutherland relation (not just Stokes-Einstein: Sutherland's work preceded Einstein's and was the more comprehensive study) is also applied. As is usual, (ref. 44), the "fractional" form is required. Again this empirical analysis should only be applied above twice the critical density. It is unclear whether the data displayed in supplementary Fig 1 meet this criterion. It would be more sensible to plot $\ln(D/T)$ versus $\ln(1/\eta)$ isotherms directly so that one could better see any temperature dependence (when the literature data are included, as in the figure. One must also take care to use reliable high-pressure viscosity data for methane, which are surprisingly scarce (e.g. Ref. 22 has data for only one isotherm, at 25 oC; see Friend et al. J. Phys. Chem. Ref. Data 18, 583 (1989)). I think the analysis needs more careful consideration.

The authors suggest that the product $(D\eta/\rho)$ is a constant for their "room-temperature" isotherm and speculate that this might be a general relationship. They do not, however, test this hypothesis further with the literature self-diffusion and viscosity data for methane they employ in the SES analysis. The hypothesis that $(D\eta/\rho)$ is a constant for a given fluid is a very old one (e.g. Dullien, Amer Inst Chem Eng. 18, 62 (1972)) and has been examined many times over the years, with increasingly more sophisticated variants (Tyrrell and Harris, Diffusion in Liquids, Butterworths, 1984, p. 295 ff; Boned et al. Phys. Rev. E 69, 031203 (2004); Suarez-Iglesias et al. Fluid Phase Equil. 269, 80 (2008)). So there is nothing novel here.

The authors have not considered comparison with the very important Lennard-Jones fluid for which there are two good sets of high-pressure molecular dynamics results over a broad range of thermodynamic states [Meier et al., J. Chem. Phys. 121, 3671 (2004); Baidakov et al., J. Chem. Phys. 137, 164507 (2012)]. This might be more realistic than the hard-sphere model. One can scale with the relevant critical parameters.

It would also be interesting to see if, when combined with the NMR data, thermodynamic (density) scaling applies at these very high densities (though the temperature would have to be known! See below). If the data are at all reliable, such scaling should be possible. (E. R. López, A. S. Pensado, M. J. P. Comuñas, A. A. H. Pádua, J. Fernández, and K. R. Harris, J. Chem. Phys. 134, 144507 (2011).).

- Will the work be of significance to the field and related fields? How does it compare to the established literature?

I think data for a single isotherm is insufficient to give a paper of general significance. Though the data are of value due to the very high pressures reached, the conclusions drawn by the authors do not extend our knowledge beyond what is already known. There is no attempt to link the results to the points made in the introduction regarding the importance of investigating the self-diffusion of methane (or its transport properties as a group) or how such a study can be applied, e.g., to planetary physics.

- Does the work support the conclusions and claims, or is additional evidence needed?

There is one glaring omission – the temperature at which the measurements were made. “Room temperature” has no meaning: is it 15 °C, 25 °C, 35 °C, or some other value? Was the cell at the same temperature as the laboratory? One could imagine that it might be heated by the surrounding apparatus. It could have been measured quite simply. This is important in making comparisons with the other experimental studies cited, all of which specify not only the temperatures, but their uncertainties. It is also important for the calculation of densities and viscosities used in the data analysis.

- Are there any flaws in the data analysis, interpretation and conclusions? - Do these prohibit publication or require revision?

- Is the methodology sound? Does the work meet the expected standards in your field?

The lack of a temperature measurement means that this work could not be published in most physical chemistry or chemical physics journals, especially those specializing in thermo-physical properties.

- Is there enough detail provided in the methods for the work to be reproduced?

I am not familiar enough with the nuts and bolts application of QENS experiments to be able to respond to this query.

Other matters:

- Equation 2 uses 5 parameters to fit 10 data points (Supplementary Table 1). This seems excessive. The data show a simple cubic dependence of $\ln(D)$ on p , in common with many other substances, so

there seems to be no need for the 4th term in eq. 2. Such a cubic dependence (for viscosity) was first reported by Bridgman in the 1920s.

- The standard deviation is 3%, which seems quite good for such work, being done without any temperature control. (See attached file). Regrettably, no standard deviation was given by the authors: usually this is mandatory when reporting such fits.
- The normally used SI unit for diffusion coefficients is m^2s^{-1} . I suggest the authors use the standard IUPAC representation for physico-chemical properties, that is, number = physical quantity/unit, in tables and on graph axis labels.
- The discussion of self-diffusion coefficient values on pp. 6 and 7 under different conditions seems unnecessary, being not to the point.
- There are 4 sets of NMR high pressure data cited (refs 17, 18, 19 and 20) but only 3 are discussed on p. 7, ref. 20 being omitted. The latter set of data extend to 0.22 GPa at 140 K; those of ref. 17 extend to 0.17 GPa at 223 K, not 0.16 GPa as stated. It is more helpful to compare data as a function of density, (or volume), rather than pressure. The hard-sphere theory comparison then follows naturally.
- There is a factual error on p. 8 regarding the discussion of self-diffusion of alcohols at high pressure. Harris (ref. 36) reported a linear molar volume dependence for \sqrt{D} , not D . Alcohols do not fit the Stokes-Einstein relation due to H-bonding.
- I obtain a good FSE fit for the authors' results, obtained by standard regression, with $\xi = 0.73$, a standard deviation of 5% and a maximum deviation of 7% (see attached file). I think the authors' calculations are slightly in error, as they give $\xi = 0.76$, no standard error (this should always be given for such fits) and a maximum deviation of 12%.

Conclusion

Overall, I do not think this paper is suitable for Nature Communications, as the work is neither of broad, general interest nor particularly novel. The data – once the relevant temperature is given – would be of interest to fluid/liquid state chemical physicists, and a revised paper would be better published in the Journal of Chemical Physics, or a similar journal. Any revision should fully encompass both the new work and previous data for dense fluid methane in the data analysis.

Reviewer #2 (Remarks to the Author):

The authors report new measurements of methane's translational diffusion coefficient from 1.2 to 14 kbar at ambient temperature, using quasi-elastic neutron scattering (QENS). The latter pressure corresponds approximately to the freezing pressure at ambient temperature. The data is important and of good quality, and will be of interest to planetary and liquid-state scientists.

The data analysis reveals breakdown of Stokes-Einstein behavior, and the authors propose a phenomenological scaling relationship that describes their data well, namely constancy of the quantity (translational diffusion coefficient \times shear viscosity/density.) The generality of this behavior remains to be tested across a range of temperatures, substances and pressures.

Overall, this is a significant contribution that extends the range of pressures for which methane diffusivity data at ambient temperature is available. It deserves eventual publication in Nature Communications, but the authors should first address the points listed below.

1. No precise information is provided about the temperature at which the experiments were performed: instead, the terminology "ambient temperature" is used throughout. The authors should explain why more precise information is not provided. Was it not possible to thermostat the high-pressure cells? Why? Can they provide an estimate of the range of "ambient temperatures" under which the experiments were conducted? They should also provide an estimate of the errors incurred by not specifying the temperature. These discussions should be included in the body of the paper, not relegated to the supplementary information.

2. The authors say that methane's freezing pressure at ambient temperature is 1.4 GPa. However, they also report in the Results section that no freezing was observed at 1.44 GPa. These two statements are contradictory, and the authors should provide an explanation.

3. Figure 4 shows that the data can be described well by the phenomenological rough hard sphere theory. This involves regressing a "coupling parameter" and an effective diameter (here a density-dependent diameter) from a comparison between (smooth) hard sphere simulation data and experiment. The fact that the data can be represented well by a rough hard sphere approach implies

that there is rotational-translational coupling. However, in extracting diffusivity values from the QENS data, the authors invoke a diffusion model that ignores such coupling. The authors should address this inconsistency.

Reviewer #3 (Remarks to the Author):

Among all the liquids/gases relevant to our life, methane is perhaps the second most important one, following water. It is the chemical we use to heat up our house, drive some of our efficient vehicles, produce fertilizer, and even fuel rockets to space. Outside earth, methane also has critical planetary physical consequences. However, our knowledge of the physical properties of methane under extreme conditions that are of astrophysical and geological relevance is still quite limited. In this paper, Bove et al. performed beautiful high-pressure QENS measurements to determine the translational diffusion coefficient of supercritical methane. These high-pressure measurements of inflammable and explosive materials are very challenging, not to mention on a neutron beam line. I praise for the excellent work of the team. The data analysis using Lorentzian fitting and the jump random diffusion model is straightforward. I strongly recommend the publication of this paper on Nature Communications.

Having said so, I do have a few questions/suggestions on their data interpretation for the authors.

1. In the inset of Figure 3, the authors pointed out a deviation from the linear dependence of the self-diffusion coefficient on the molar volume at high pressures. However, the deviation is not significant. Although the deviation is beyond the error bars, the error bars obtained from the data fitting can be easily biased by the specific model and fitting procedures applied. What is not captured by the error bar is the systematic deviation brought by the QENS model. We can also draw one straight line to fit the entire pressure range with confidence. This is not a significant point of the paper. However, such a claim may potentially bite the authors back in the future.

2. I'm not convinced by the interpretation of the apparent fractional SER. The authors have already shown the effective hydrodynamic diameter depends on pressure, and thus indirectly depends on the viscosity. Hence, a simple log-log plot of D vs η cannot be used to test the SER. Honestly, I wouldn't read too much into it at all. On the other hand, immediately after, the authors show that $D\eta/\rho$ is basically a constant. This is a much stronger evidence of the pressure (and thus viscosity) dependent hydrodynamic diameter. I personally would suggest removing Figure 5 and replacing it

with Figure S2. There is no need to over-interpret. The data itself is already significant and warrant a publication.

3. A minor comment and suggestion for future work: room temperature is far above the critical point of methane (190.6 K). Are the authors planning to perform more measurements at lower temperatures and near the critical point? It would be very interesting to see the critical slowing down. On the other hand, although the measurements presented in this paper are at RT. What is the pressure of the critical isochore at this temperature? Are there any remote evidence of critical fluctuations (or Widom line as many advertise)?

Y Z

Associate Professor, Donald Biggar Willett Faculty Scholar

Department of Nuclear, Plasma, and Radiological Engineering

Department of Electrical and Computer Engineering

Program of Computational Science and Engineering

Center for Biophysics and Quantitative Biology

Beckman Institute for Advanced Science and Technology

University of Illinois at Urbana-Champaign

<http://z.engineering.illinois.edu/>

Reviewer #1 (Remarks to the Author):

- What are the noteworthy results?

This work reports QENS measurements of the self-diffusion coefficient of methane in the super-critical region along a single isotherm, extending to the freezing line, that is to 1.44GPa. The work complements and extends the range of thermodynamic states of previous high pressure measurements, made mainly by spin echo NMR, for the liquid and dense gas states. The liquid state measurements also extend to the freezing line.

The data are analysed using several commonly used theoretical and empirical approaches: a) smooth and rough hard sphere theory, that is a comparison with literature molecular dynamics simulations for the hard-sphere fluid, whereby a temperature dependent diameter and an empirical “translational-rotational” coupling parameter can be assigned, b) the Stokes-Einstein-Sutherland equation relating the self-diffusion coefficient to the viscosity, which in this context is a comparison of the density or pressure dependence of the self-diffusion coefficient with that of the viscosity.

We thank the reviewer for the many useful comments/suggestions and for bringing to our attention relevant publications. In the revised version of the manuscript, changes addressing the concerns of reviewer #1 are shown in brown.

The authors conclude from the hard sphere comparison that:

a) Methane does not fit the smooth-hard sphere model, confirming what is known from earlier studies (ref. 41). The deviation from a linear dependence of D on the molar volume shown in Fig 3 is actually the norm for molecular fluids, being first reported by Parkhurst and Jonas in the 1970's (ref. 49). Though this is the first clear observation of such behaviour for methane, Harris and Trappeniers (ref 20) reported a similar trend for their highest pressure data (their Fig. 4). Van der Gulik has also reported similar behaviour for the viscosity of liquid CO₂ (maximum pressure 0.4 GPa) (P S van der Gulik, Physica A, 23, 81 (1997)).

The referee remarks that the deviation from a linear dependence of D on the molar volume shown in our (old) Fig 3 is the norm for molecular fluids and that similar trends were reported before, also for methane. However, our deviation is markedly larger than that reported in the literature because we go to much higher pressures.

In order to include literature data in a plot showing the smooth-hard sphere model failure, we plotted the relevant quantity D/\sqrt{T} versus the molar volume, following Harris et Trappeniers 1980 (ref 20 of the main text), including all of the available literature isotherms:

As can be readily seen from the figure, the slope of our high-density (low-molar volume) data represents a significant deviation from the slope of the literature isotherms. However, as we find that this figure and the related discussion would be difficult to follow for a general reader, we decided to remove the plot of D as a function of molar volume from the manuscript (the inset of Fig 3). Instead, we now plot the reduced diffusion coefficient defined by Dymond [Physica 75, 100–114 (1974)] as a function of reduced number density in Supplementary Fig 3, hereafter reported. This kind of approach also follows previous analysis of diffusivity data of compressed methane [Harris and Trappeniers 1980].

Supplementary Fig 3 seems to us a clearer way of presenting the same finding. Furthermore, relevant literature isotherms could be included without compromising the readiness of the figure. Supplementary Fig 3 also reports the molecular dynamics simulation results (ref 46 of the main text) as a full line, so the deviation from the hard-

sphere behaviour is clearly visible. Supplementary Fig 3 clearly shows that the deviation of our data at 300 K from the smooth hard-sphere behaviour is larger than that observed for liquid methane in ref 20 of the main text (ref 14 of the Supplementary Information).

We thus agree with the referee on the comment that the present study is the first clear experimental observation of departure from the smooth hard-sphere model for methane, while ref 20 reported a similar trend for their few highest-pressure points. In the revised manuscript, this point is now mentioned at page 11 and discussed in details in Supplementary Note 3.

b) Instead, the data can be fitted with the use of a translational-rotational coupling factor, but only with the use of a density dependent diameter. Again this is a result of the non-linear molar volume dependence, or, more precisely, the deviation from the smooth hard spheres volume dependence, which only approximates a linear molar volume dependence.

Such comparisons can only be made between twice the critical density, 0.32 g cm⁻³ for CH₄, (see ref. 39) and the freezing density of the hard sphere fluid, 0.94. The data presented do fit between these limits, but this point is omitted by the authors.

We agree with the referee on this point. Our pressure range is indeed between twice the critical density and the freezing density of the hard sphere fluid but we forgot to mention this in the manuscript. At page 4 of the revised manuscript, we now mention that the density at the lowest investigated pressure is more than twice the density at the critical point. At page 9, we remind the reader that Chandler's model is only valid between twice the critical density and the freezing density of the hard sphere fluid. At page 10 of the Supplementary Information, we explain that the reduced number density is about 0.93 at our highest pressure point for the sigma value that we deduced at this pressure (3.43 Ang.).

The Stokes-Einstein-Sutherland relation (not just Stokes-Einstein: Sutherland's work preceded Einstein's and was the more comprehensive study) is also applied. As is usual, (ref. 44), the "fractional" form is required. Again this empirical analysis should only be applied above twice the critical density. It is unclear whether the data displayed in supplementary Fig 1 meet this criterion. It would be more sensible to plot $\ln(D/T)$ versus $\ln(1/\eta)$ isotherms directly so that one could better see any temperature dependence (when the literature data are included, as in the figure. One must also take care to use reliable high-pressure viscosity data for methane, which are surprisingly scarce (e.g. Ref. 22 has data for only one isotherm, at 25 oC; see Friend et al. J. Phys. Chem. Ref. Data 18, 583 (1989)). I think the analysis needs more careful consideration.

We added a comment about the fact that the SES equation should not be expected to work below twice the critical density (page 14).

The reviewer suggested to plot $\ln(D/T)$ versus $\ln(1/\eta)$ instead of $\ln(D)$ versus $\ln(\eta)$ in Fig 5. We made this change. This is a minor change since our data and the literature data reported in Fig 5 are at essentially same temperature (the maximum difference is 5 K). We also changed everywhere in the text "Stokes-Einstein" relation with "Stokes-Einstein-Sutherland" relation.

We prepared a plot of $\ln(D/T)$ versus $\ln(1/\eta)$ including some low-T isotherms. This figure is shown below:

The figure above shows that the slope of $\ln(D/T)$ versus $\ln(1/\eta)$ is close to 1 for the low-T isotherms therefore the SES equation holds in compressed liquid methane at low temperatures (110, 140, and 160 K). However, this figure is hardly readable if further isotherms are included, such as the room-temperature isotherms. We thus decided to plot the room-temperature isotherms only in Fig 5 and we created a new figure (Supplementary Fig 5) where we plot $D\eta$ versus pressure for literature data along the low-T isotherms.

It is clear from Supplementary Fig 5 that the SES equation holds in compressed liquid methane at low temperature. This observation is now mentioned in the revised manuscript (page 14). However, it must be stressed that the viscosity values we used at low temperatures are predicted values from ref 24, not experimental ones. Unfortunately, no experimental viscosity data exist at the relevant T and p.

As mentioned by the reviewer, ref 22 has data for only one isotherm. However, the same group published viscosity data for the 273 K isotherm [van der Gulik et al. *Fluid Phase Equilibria*, 79 (1992) 301-311]. The review paper mentioned by the reviewer [Friend et al. *J. Phys. Chem. Ref. Data* 18, 583 (1989)] compares viscosity data for methane at very modest pressures (up to 0.055 GPa). We added a sentence about the lack of high-pressure, low-temperature viscosity data of methane at the end of the Discussion section (page 19).

The authors suggest that the product $(D\eta/\rho)$ is a constant for their “room-temperature” isotherm and speculate that this might be a general relationship. They do not, however, test this hypothesis further with the literature self-diffusion and viscosity data for methane they employ in the SES analysis. The hypothesis that $(D\eta/\rho)$ is a constant for a given fluid is a very old one (e.g. Dullien, *Amer Inst Chem Eng.* 18, 62 (1972)) and has been examined many times over the years, with increasingly more sophisticated variants (Tyrrell and Harris, *Diffusion in*

Liquids, Butterworths, 1984, p. 295 ff; Boned et al. Phys. Rev. E 69, 031203 (2004); Suarez-Iglesias et al. Fluid Phase Equil. 269, 80 (2008)). So there is nothing novel here.

We had already tested this hypothesis ($D\eta/\rho = \text{const}$) on the room-temperature literature self-diffusion and viscosity data (the same data we employed in the aforementioned SES analysis). This test was not discussed in the main text but it was shown in the old Fig S2. Hence, we kept the same figure in the Supplementary Information (now Supplementary Fig 6) and added a sentence about the literature data in the main text (page 15). We also prepared a new figure (Supplementary Fig 7) where we test the same hypothesis ($D\eta/\rho = \text{const}$) on the literature data along low-T isotherms.

We also added a sentence about the work by Dullien (page 14-15) and a sentence about the more sophisticated variants (page 15).

The authors have not considered comparison with the very important Lennard-Jones fluid for which there are two good sets of high-pressure molecular dynamics results over a broad range of thermodynamic states [Meier et al., J. Chem. Phys. 121, 3671 (2004); Baidakov et al., J. Chem. Phys. 137, 164507 (2012)]. This might be more realistic than the hard-sphere model. One can scale with the relevant critical parameters.

In the previous version of the manuscript, we compared our low-pressure self-diffusion coefficient with the prediction of a molecular dynamics simulation study of methane using a Lennard-Jones potential with electrostatic interactions (now page 8). We also indirectly compared our results with the Lennard-Jones model when we compare the exponent of the fractional SES equation fitting our data and that reported in the literature for the Lennard-Jones fluid (now page 18).

However, it is true that we were missing a comparison with the pressure dependence of the diffusion coefficient predicted for the Lennard-Jones fluid. In the revised version of the manuscript, we included such comparison using the results of ref [Baidakov et al., Fluid Ph. Equilibria 305, 106-113 (2011)]. We now comment this comparison at page 8 and report a plot in Supplementary Fig 2, also reported hereafter.

We find a good agreement at our two lowest pressure points for the Lennard-Jones parameters ϵ and σ taken from the literature of methane ($\epsilon/k_B = 150$ K and $\sigma = 3.8$ Å). However, as pressure increases above 0.2 GPa, our data deviate significantly from the Lennard-Jones prediction.

It would also be interesting to see if, when combined with the NMR data, thermodynamic (density) scaling applies at these very high densities (though the temperature would have to be known! See below). If the data are at all reliable, such scaling should be possible. (E. R. López, A. S. Pensado, M. J. P. Comuñas, A. A. H. Pádua, J. Fernández, and K. R. Harris, *J. Chem. Phys.* 134, 144507 (2011).).

If we understand correctly, the reviewer suggests to plot D as a function of $\rho^\gamma T^{-1}$ for our data and the literature data of methane. As mentioned above, we see a different slope for the volume and number density dependences of our data compared to those of the data available in the literature at different temperatures and lower pressures. So we expect that the same difference would be visible in a plot of D as a function of $\rho^\gamma T^{-1}$ and that very little could be learnt from this analysis.

Moreover, we are not sure if such a thermodynamic density scaling has ever been applied to diffusivity data of methane, as we are not aware of any analysis of this type in the literature.

- Will the work be of significance to the field and related fields? How does it compare to the established literature?

I think data for a single isotherm is insufficient to give a paper of general significance. Though the data are of value due to the very high pressures reached, the conclusions drawn by the authors do not extend our knowledge beyond what is already known. There is no attempt to link the results to the points made in the introduction regarding the importance of investigating the self-diffusion of methane (or its transport properties as a group) or how such a study can be applied, e.g., to planetary physics.

The very challenging experiments reported here allowed us to explore the diffusion mechanism of dense fluid methane in an unprecedentedly wide pressure variation range along an isotherm and to compare the result with standard models for simple fluids. The measurement of the diffusion coefficient itself and the conclusions we drawn on the failure of standard models in depicting the diffusive behaviour of high-density

supercritical methane seem to us a sufficient justification for the publication of this paper. We are confident that the present work will stimulate further experiments and simulations at different temperatures and/or in similar systems. The focus of this work being to study the effect of pressure on the self-diffusion coefficient, we do not think that a few additional isotherms would considerably change the physical conclusions in the present manuscript.

We think that there is a general lack of high-quality measurements of transport properties of simple molecular fluids like methane under extreme conditions. This is the reason why the smooth and rough hard-sphere models, as well as the Stokes-Einstein-Sutherland relation could never be tested experimentally neither on supercritical methane, nor on a dense supercritical simple fluid in general. Reviewers#2 and #3 also think that our results are of sufficient general significance to deserve publication in Nature Communications.

As concerns linking our results to the points made in the introduction, we discuss the possible orientational ordering of the methane molecules at room temperature and pressures above 0.8 GPa at page 17. We are not able to solve this matter but we did attempt to link our results to this open scientific problem. However, we feel that applying our results to other research fields would fall beyond the scope of the present work. Instead, in the revised version of the manuscript, we provide some ideas about the possible ways our findings could be useful, both in the industry and in Earth science (page 19-20). We also briefly discuss recent developments in the field of compressed supercritical fluids following a comment by reviewer #3 (namely, regarding the so-called Widom and Frenkel lines) and identify some needs to be addressed by possible future studies. We added this paragraph at the end of the Discussion section:

The present study might be extended to fluid methane along lower- T isotherms in the vicinity of the critical point as well as along higher- T isotherms, where higher pressures and densities can be achieved. There is also a need for more high-pressure viscosity data of methane at temperatures below 273 K, the current limit being 0.031 GPa in the experimental literature dataset[Friend1989] and 0.2 GPa from the equation of state-like viscosity model of ref. [Younglove1987]. In general, data on the properties of compressed supercritical fluids are surprisingly scarce, despite the increasing number of applications of supercritical fluids in extraction and purification technologies, for example in food, petrochemical, nuclear waste, and pharmaceutical industries. The solubility of supercritical fluids depends on density and diffusivity. A better knowledge of the diffusivity of supercritical fluids would allow for an optimization of their dissolving ability by tuning pressure and temperature. We believe that technology would benefit from such a guidance. We must acknowledge that significant progresses have been made in recent years concerning the fundamental problem of dividing the phase diagram into separate regions where a supercritical fluid possesses liquid-like and gas-like properties[Bolmatov2013,Smith2017]. There is still a lot to do on compressed fluid binary mixtures. Pressures of the order of a fraction of GPa are increasingly accessible by the industry and are also relevant for Earth and planetary science. As an example, methane-rich hydrous fluids are ubiquitous in the Earth, where they exist down to upper mantle conditions, and are believed to have a strong impact on elastic waves propagation at depth [Shirey2013].

- Does the work support the conclusions and claims, or is additional evidence needed?

The is one glaring omission – the temperature at which the measurements were made. “Room temperature” has no meaning: is it 15 oC, 25 oC, 35 oC, or some other value? Was the cell at the same temperature as the laboratory? One could imagine that it might be heated by the surrounding apparatus. It could have been measured quite simply. This is important in making

comparisons with the other experimental studies cited, all of which specify not only the temperatures, but their uncertainties. It is also important for the calculation of densities and viscosities used in the data analysis.

We apologize for this oversight. We did measure the temperature close to the sample all along the experiment and found it is 300 ± 1 K. This has been clarified in the revised version of the manuscript. Now we also specify the temperature of the literature “room-temperature” measurements of D and of the simulations of D , both in the text and in the figures.

- Are there any flaws in the data analysis, interpretation and conclusions? - Do these prohibit publication or require revision?

- Is the methodology sound? Does the work meet the expected standards in your field?

The lack of a temperature measurement means that this work could not be published in most physical chemistry or chemical physics journals, especially those specializing in thermo-physical properties.

We answered this point above.

- Is there enough detail provided in the methods for the work to be reproduced?

I am not familiar enough with the nuts and bolts application of QENS experiments to be able to respond to this query.

Other matters:

- Equation 2 uses 5 parameters to fit 10 data points (Supplementary Table 1). This seems excessive. The data show a simple cubic dependence of $\ln(D)$ on p , in common with many other substances, so there seems to be no need for the 4th term in eq. 2. Such a cubic dependence (for viscosity) was first reported by Bridgman in the 1920s.

We accepted the suggestion of the reviewer and changed the fit of Fig 3. However, a cubic dependence of $\ln(D)$ on p is certainly not the only possible option. A modified Andrade equation, a modified Vogel, Fulcher, Tammann equation, and the Woolf-Watts equations are examples of more complicated possible choices [Supplementary Material of Harris, Journal of Chemical Physics, 131 (2009)]. In particular, the modified Andrade equation has been used for methane [Supplementary Material of Harris, Journal of Chemical Physics, 131 (2009)]. At high pressures, a linear dependence of the viscosity on pressure has been also widely discussed [van der Gulik, Physica A 256 (1998) 39-56] and constitutes an example of a simpler option.

- The standard deviation is 3%, which seems quite good for such work, being done without any temperature control. (See attached file). Regrettably, no standard deviation was given by the authors: usually this is mandatory when reporting such fits.

We included the value of the standard deviation in the revised manuscript (page 7).

- The normally used SI unit for diffusion coefficients is m^2s^{-1} . I suggest the authors use the standard IUPAC representation for physico-chemical properties, that is, number = physical quantity/unit, in tables and on graph axis labels.

The unit cm^2s^{-1} is commonly used for diffusion coefficients in the literature. However, we agreed to change it into m^2s^{-1} everywhere in the revised manuscript. We also adopted the standard IUPAC representation in tables and on graph axis labels.

- The discussion of self-diffusion coefficient values on pp. 6 and 7 under different conditions seems unnecessary, being not to the point.

The reviewer is probably referring to the comparison with the diffusion coefficient of liquid methane at ambient pressure and with that of liquid methane along the melting line. Those are certainly not the main points of the paper but seem pertinent to us. We shortened the paragraph but decided to leave the two comparisons (page 7).

- There are 4 sets of NMR high pressure data cited (refs 17, 18, 19 and 20) but only 3 are discussed on p. 7, ref. 20 being omitted. The latter set of data extend to 0.22 GPa at 140 K; those of ref. 17 extend to 0.17 GPa at 223 K, not 0.16 GPa as stated. It is more helpful to compare data as a function of density, (or volume), rather than pressure. The hard-sphere theory comparison then follows naturally.

The importance of the comparison with the literature data at different temperatures has been increased in the first subsection of the Results section and in the Supplementary Information, with four new figures reporting literature data at T different from room temperature. Accordingly, the title of the subsection has been modified from “Fitting model and obtained diffusion coefficient” into “Fitting model, obtained diffusion coefficient and comparison with the literature data”. In particular, we added a sentence about the pressure and temperature ranges of the non-room temperature literature diffusivity data (page 8) and a phase diagram (Supplementary Fig 1) where literature high-pressure diffusivity data are located at their temperatures and pressures.

Fitting $\ln(D)$ in terms of the pressure (see comment of the reviewer above) requires that we plot D or $\ln(D)$ as function of pressure. We think this is also helpful for the many readers who might be interested in the pressure dependence of the diffusion coefficients more than in its density (or volume) dependence. However, we added the density on the top horizontal axis of Fig 3.

- There is a factual error on p. 8 regarding the discussion of self-diffusion of alcohols at high pressure. Harris (ref. 36) reported a linear molar volume dependence for ηD , not D . Alcohols do not fit the Stokes-Einstein relation due to H-bonding.

Thanks! We amended this error.

- I obtain a good FSE fit for the authors' results, obtained by standard regression, with $\xi = 0.73$, a standard deviation of 5% and a maximum deviation of 7% (see attached file). I think the authors' calculations are slightly in error, as they give $\xi = 0.76$, no standard error (this should always be given for such fits) and a maximum deviation of 12%.

We thank the reviewer for checking and letting us know. We corrected this and indicated the value of the standard deviation (page 14). The difference in the fitted value of ξ came from the fact that we were fitting D as function of η directly in old Fig 5.

Conclusion

Overall, I do not think this paper is suitable for Nature Communications, as the work is neither of broad, general interest nor particularly novel. The data – once the relevant temperature is given – would be of interest to fluid/liquid state chemical physicists, and a revised paper would be better published in the Journal of Chemical Physics, or a similar journal. Any revision should fully encompass both the new work and previous data for dense fluid methane in the data analysis.

For the aforementioned reasons, we disagree with the conclusion of the reviewer.

Reviewer #2 (Remarks to the Author):

The authors report new measurements of methane's translational diffusion coefficient from 1.2 to 14 kbar at ambient temperature, using quasi-elastic neutron scattering (QENS). The latter pressure corresponds approximately to the freezing pressure at ambient temperature. The data is important and of good quality, and will be of interest to planetary and liquid-state scientists.

The data analysis reveals breakdown of Stokes-Einstein behavior, and the authors propose a phenomenological scaling relationship that describes their data well, namely constancy of the quantity (translational diffusion coefficient \times shear viscosity/density.) The generality of this behavior remains to be tested across a range of temperatures, substances and pressures. Overall, this is a significant contribution that extends the range of pressures for which methane diffusivity data at ambient temperature is available. It deserves eventual publication in Nature Communications, but the authors should first address the points listed below.

We thank reviewer #2 for appreciating our work and for the useful comments/suggestions, to which we answer below. In the revised version of the manuscript, changes addressing the concerns of reviewer #2 are shown in violet.

1. No precise information is provided about the temperature at which the experiments were performed: instead, the terminology “ambient temperature” is used throughout. The authors should explain why more precise information is not provided. Was it not possible to thermostat the high-pressure cells? Why? Can they provide an estimate of the range of “ambient temperatures” under which the experiments were conducted? They should also provide an estimate of the errors incurred by not specifying the temperature. These discussions should be included in the body of the paper, not relegated to the supplementary information.

We apologize for this oversight. We did measure the temperature close to the sample all along the experiment and found it is 300 ± 1 K. This has been clarified in the revised version of the manuscript.

2. The authors say that methane's freezing pressure at ambient temperature is 1.4 GPa. However, they also report in the Results section that no freezing was observed at 1.44 GPa. These two statements are contradictory, and the authors should provide an explanation.

The freezing pressure of methane is 1.38 ± 0.02 GPa at 300 K [E. H. Abramson, Melting curves of argon and methane High Pressure Research, 31:4, 549-554]. We apologize for

omitting the error bar in the previous version of the manuscript. No freezing was observed in our measurement at 1.44 ± 0.05 GPa. So, when the error bars are considered, no inconsistency is found. Still, similar to the case of undercooling, metastability is not uncommon in fluids compressed above their freezing point. This is now mentioned in the revised manuscript at page 5.

Interestingly, the molecular dynamics simulation paper we compare our results to (ref 46) reported results in the metastable fluid region above the freezing density too (up to reduced number densities as high as 1.01).

3. Figure 4 shows that the data can be described well by the phenomenological rough hard sphere theory. This involves regressing a “coupling parameter” and an effective diameter (here a density-dependent diameter) from a comparison between (smooth) hard sphere simulation data and experiment. The fact that the data can be represented well by a rough hard sphere approach implies that there is rotational-translational coupling. However, in extracting diffusivity values from the QENS data, the authors invoke a diffusion model that ignores such coupling. The authors should address this inconsistency.

Yes, in extracting diffusivity values from the QENS data, we had to use a diffusion model that ignores the rotational-translational coupling. Such an approximation is very commonly made, and it is actually essential to obtain an analytical expression for the dynamic structure factor. We explain more in details this choice below.

First, this approximation is commonly made in contexts where both translational and rotational contributions are needed to fit the data [for example L. E. Bove et al., Phys. Rev. Lett. 111, 185901 (2013); U. Ranieri et al., J. Phys. Chem. B 120, 9051–9059 (2016)]. Here, however, only the translational contribution is needed to fit the data because the rotational motion is out of the investigated timescale window, which makes the rotational-translational coupling less pertinent than in other published studies. In general, the rotational-translational coupling is more important at low temperatures and here the temperature of the experiments was significantly above the melting temperature of methane at ambient pressure. The fact that the translational contribution only enters the energy window of the experiment justifies to omit the rotational term and the second order roto-translational term for the modelling of the measured spectra.

Second, the reviewer correctly says that a value of A smaller than unity was needed for the rough hard sphere model to fit the data. However, the rotational-translational coupling parameter used in Figs 4c and 4d is $A=0.927$, which indicates a fairly small rotational-translational coupling (in this model, $A=1$ is the value associated with no rotational-translational coupling). The value of 0.927 is not deduced from our data but it is taken from low-pressure literature analysis (ref 42). It is likely that stronger rotational-translational coupling exists at high pressures, as we mention at page 3 of the Supplementary Material. Yet, a Lorentzian function continues to correctly fit our experimental spectra up to the highest pressures over a narrow fitting range close to zero energy transfer, as can be seen in Fig 1.

In summary, this study used a good energy resolution to obtain the translational diffusion coefficient of supercritical methane at high pressure. The timescale of the translational diffusion matches very well the instrumental setup when IN6 is set to have a wavelength of the incoming neutrons of 5.12 Å. In time-of-flight neutron scattering, the natural drawback of a good energy resolution is a limited energy transfer range. So, it is not surprising that the present study cannot make conclusions on the rotational contributions (which is faster than the translational contribution) and the rotational-translational coupling. Since the fits of the spectra are excellent, we see no reason to believe that our estimation of the translational diffusion coefficient could be erroneous.

Reviewer #3 (Remarks to the Author):

Among all the liquids/gases relevant to our life, methane is perhaps the second most important one, following water. It is the chemical we use to heat up our house, drive some of our efficient vehicles, produce fertilizer, and even fuel rockets to space. Outside earth, methane also has critical planetary physical consequences. However, our knowledge of the physical properties of methane under extreme conditions that are of astrophysical and geological relevance is still quite limited. In this paper, Bove et al. performed beautiful high-pressure QENS measurements to determine the translational diffusion coefficient of supercritical methane. These high-pressure measurements of inflammable and explosive materials are very challenging, not to mention on a neutron beam line. I praise for the excellent work of the team. The data analysis using Lorentzian fitting and the jump random diffusion model is straightforward. I strongly recommend the publication of this paper on Nature Communications.

We would like to thank the reviewer for strongly recommending the publication of our paper in Nature Communications and for the useful comments/suggestions, to which we answer below. In the revised version of the manuscript, changes addressing the concerns of reviewer #3 are shown in red.

Having said so, I do have a few questions/suggestions on their data interpretation for the authors.

1. In the inset of Figure 3, the authors pointed out a deviation from the linear dependence of the self-diffusion coefficient on the molar volume at high pressures. However, the deviation is not significant. Although the deviation is beyond the error bars, the error bars obtained from the data fitting can be easily biased by the specific model and fitting procedures applied. What is not captured by the error bar is the systematic deviation brought by the QENS model. We can also draw one straight line to fit the entire pressure range with confidence. This is not a significant point of the paper. However, such a claim may potentially bite the authors back in the future.

The deviation seems significant to us and is closely related to the deviation from the hard sphere model (see the comments of reviewer #1 and our reply). However, we decided to remove the plot of the self-diffusion coefficient as a function of molar volume. The same piece of information is now available from the plot of the reduced diffusion coefficient as a function of reduced number density (Supplementary Fig 3), where no fit is needed. Moreover, literature data along other isotherms and the computational prediction for the hard-sphere fluid are also reported in Supplementary Fig 3.

We agree that the plot of the self-diffusion coefficient as a function of molar volume was not a significant point of the paper, since the self-diffusion coefficient was (and is) also plotted as a function of density in the manuscript (Fig 4). Fig 4 and Supplementary Fig 3 seem better figures for showing the deviation from the hard sphere model. The inset of Figure 3 was probably unnecessary.

2. I'm not convinced by the interpretation of the apparent fractional SER. The authors have already shown the effective hydrodynamic diameter depends on pressure, and thus indirectly depends on the viscosity. Hence, a simple log-log plot of D vs η cannot be used to test the SER. Honestly, I wouldn't read too much into it at all. On the other hand, immediately after, the authors show that $D\eta/\rho$ is basically a constant. This is a much stronger evidence of the pressure (and thus viscosity) dependent hydrodynamic diameter. I personally would suggest removing Figure 5 and replacing it with Figure S2. There is no need to over-interpret. The data itself is already significant and warrant a publication.

The reviewer is right: We show that the effective hard sphere diameter depends on pressure in the subsection about the hard sphere model and this suggests that the SE relation cannot be fulfilled in its standard form, with a constant hydrodynamic diameter. The two "diameters" do not have exactly the same meaning, but we do agree with the reviewer. We added the following sentence to page 13: "*Vice versa, if σ is known to depend on the thermodynamic conditions (as we have shown in the previous subsection) then the SES relation is expected to fail.*"

On the other hand, violation of the fractional SE relation does not obviously follow. This is why the figure about the SE relation (Supplementary Fig 4) is in the Supplementary Information and the figure about the fractional SE relation (Fig 5) is in the main text. The fractional SE relation is commonly written as a proportionality, like in our eq. (5), where the hydrodynamic diameter does not explicitly appear. The power law exponent of the fractional SE relation is typically a free parameter and this gives the fractional SE relation enough flexibility to be fulfilled also in case of pressure-dependent diameters.

Moreover, we think that the violation of the fractional SE relation is interesting per se and deserves to be discussed in the main text. The SE and fractional SE relations are commonly used for extrapolating either the diffusion coefficient or the viscosity. If the reviewer thinks this would be better, we can use two different symbols for the two diameters (Eq 3 and Eq 4).

3. A minor comment and suggestion for future work: room temperature is far above the critical point of methane (190.6 K). Are the authors planning to perform more measurements at lower temperatures and near the critical point? It would be very interesting to see the critical slowing down. On the other hand, although the measurements presented in this paper are at RT. What is the pressure of the critical isochore at this temperature? Are there any remote evidence of critical fluctuations (or Widom line as many advertise)?

The minimum of the speed of sound is at about 0.02 GPa at 300 K [Yang et al. Phys. Rev. E 91, 012112 (2015)], which is actually out of the pressure range investigated in the present measurements. QENS studies at temperatures and pressures close to the critical

point might indeed provide very interesting insights. We thank the reviewer for this valid suggestion for future work. This comment inspired us several changes: We added a reminder about the phase diagram of methane at page 4, a brief perspective for future works at page 19, and a comment about Widom and Frenkel lines at page 19-20.

REVIEWERS' COMMENTS

Reviewer #1 (Remarks to the Author):

Given the substantive changes made to the manuscript I now recommend publication.

Reviewer #2 (Remarks to the Author):

The authors have addressed my comments, and those of the other reviewers, satisfactorily. I recommend publication of this interesting work in Nature Communications.

Reviewer #3 (Remarks to the Author):

The authors have provided convincing responses. I recommend the publication of the paper as it is.

Reviewer #1 (Remarks to the Author):

Given the substantive changes made to the manuscript I now recommend publication.

We would like to thank the reviewer for recommending the publication of our paper.

Reviewer #2 (Remarks to the Author):

The authors have addressed my comments, and those of the other reviewers, satisfactorily. I recommend publication of this interesting work in Nature Communications.

We would like to thank the reviewer for recommending the publication of our paper.

Reviewer #3 (Remarks to the Author):

The authors have provided convincing responses. I recommend the publication of the paper as it is.

We would like to thank the reviewer for recommending the publication of our paper.